# Promiscuous molecules for smarter file operations in DNA-based data storage

Kyle J. Tomek [1], Kevin Volkel[2], Elaine W. Indermaur[1], James M. Tuck [2✉] & Albert J. Keung [1✉]

DNA holds significant promise as a data storage medium due to its density, longevity, and resource and energy conservation. These advantages arise from the inherent biomolecular structure of DNA which differentiates it from conventional storage media. The unique molecular architecture of DNA storage also prompts important discussions on how data should be organized, accessed, and manipulated and what practical functionalities may be possible. Here we leverage thermodynamic tuning of biomolecular interactions to implement useful data access and organizational features. Specific sets of environmental conditions including distinct DNA concentrations and temperatures were screened for their ability to switchably access either all DNA strands encoding full image files from a GB-sized background database or subsets of those strands encoding low resolution, File Preview, versions. We demonstrate File Preview with four JPEG images and provide an argument for the substantial and practical economic benefit of this generalizable strategy to organize data.

[1] Department of Chemical and Biomolecular Engineering, North Carolina State University, Raleigh, NC, USA. [2] Department of Electrical and Computer Engineering, North Carolina State University, Raleigh, NC, USA. ✉email: jtuck@ncsu.edu; ajkeung@ncsu.edu

nformation is being generated at an accelerating pace while our means to store it are facing fundamental material, energy, environment, and space limits[1]. DNA has clear potential as a data storage medium due to its extreme density, durability, and efficient resource conservation[2–6]. Accordingly, DNA-based data storage systems up to 1 GB have been developed by harnessing the advances in DNA synthesis and sequencing, and support the plausibility of commercially viable systems in the not too distant future[7–11]. However, in addition to continuing to drive down the costs of DNA synthesis and sequencing, there are many important questions that must be addressed. Foremost among them are how data should be organized, accessed, and searched.

Organizing, accessing, and finding information constitutes a complex class of challenges. This complexity arises from how information is commonly stored in DNA-based systems: as many distinct and disordered DNA molecules free-floating in dense mutual proximity[8,9,12–16]. This has two major implications. First, an addressing system is needed that can function in a complex and information-dense molecular mixture. While the use of a physical scaffold to array the DNA would ostensibly solve this challenge, analogous to how data are addressed on conventional tape drives, this would abrogate the density advantage of DNA as the scaffold itself would occupy a disproportionate amount of space. Second, while the inclusion of metadata in the strands of DNA could facilitate search, ultimately there will be many situations in which multiple candidate files contain very similar information. For example, one might wish to retrieve a specific image of the Wright brothers and their first flight, but it would be difficult to include enough metadata to distinguish the multiple images of the Wright brothers as they all fit very similar search criteria. In addition, data stored using DNA could be maintained for generations[6] with future users only having access to a limited amount of metadata and cultural memory or knowledge. Given the costs associated with DNA retrieval and sequencing, a method to preview low-resolution versions of multiple files without needing to fully access or download all of them would be advantageous.

In previously reported DNA systems, files were organized, recognized, and accessed through specific DNA base-pair interactions with ~20 nucleotides (nt) address sequences in both PCR-based file amplifications[8,13–16] and hybridization-based separations[9–11]. However, these address sequences participate in thermodynamically driven interactions that are not cleanly all-or-none as they are for conventional electronic storage addresses[17,18]. To bypass this limitation, current DNA system architectures and encoding strategies avoid any untoward cross-interactions between addresses by setting a threshold for sequence similarity (e.g., Hamming distance, HD)[16,19,20] (Fig. 1a). These limits on the address sequence space result in a reduction in the storage capacity of systems[14,21], as well as in the amount of metadata that could be included for use in search functions. Both limitations pose significant practical barriers for this technology and restrict the engineering of more advanced and useful functions[10,11].

We hypothesize that so-called nonspecific interactions in DNA-based data storage systems, conventionally viewed as a thermodynamic hindrance, can actually be leveraged to expand file address space, increase data storage capacity, and implement in-storage functions in DNA storage systems. This hypothesis is inspired by intentional nonspecific interactions that have been leveraged for DNA editing in molecular biology (e.g., site-directed mutagenesis) and more recently in DNA storage for in-storage search[10,11,22]. Here, we develop a theoretical and experimental understanding of factors that impact DNA–DNA interactions and show that we can predictably tune molecular interactions between imperfectly matched sequences in both isolated and competitive systems. To further demonstrate this concept and its potential utility in a true data storage system, individual files are encoded into three or four distinct subsets of strands (i.e., fractions of the file) that can be differentially accessed using the same accessing primer by tuning only the PCR conditions. In this approach, a small portion of a file can be previewed as a low-resolution version of the original higher-resolution image, an operation with the closest analogy to Quick Look, a function found on modern Mac operating systems or Progressive JPEG but with fundamentally different implementation. Importantly, this function uses address sequences (i.e., primer binding sites) that would have previously been discarded due to their mutual sequence similarities, and therefore does not impact total database capacities. We successfully implement File Preview for four different image files in the presence of a randomized, nonspecific 1.5 GB background to more accurately represent a realistic DNA storage system. This approach to encoding and accessing strands harnesses the intrinsic properties of DNA and implements practical functionality while remaining cost-competitive with traditional DNA storage systems. We also anticipate that this general principle of leveraging uniquely biochemical aspects of DNA molecules could be extended to implement diverse and useful functions including encoding metadata, increasing or decreasing the stringency of a search function, and prioritizing information by differentially encoding files to increase the read efficiency of frequently versus infrequently accessed data.

## Results

**PCR stringency is thermodynamically tunable.** In PCR-based DNA storage systems, data payloads, file addresses, and PCR primers that bind those addresses have typically been designed to avoid nonspecific interactions by requiring that all primers are at least six to ten or more mismatches from all other sequences in a database (6–10+ HD) (Fig. 1a)[14,16]. To test this design criterion, we incorporated the widely used NuPACK thermodynamic model into a Monte Carlo simulation and found that a HD of greater than 10 was likely required to minimize unwanted hybridizations (Fig. 1b, black line)[14,23]. We confirmed this experimentally by measuring the percentage of successful PCRs using a primer with 10 strand addresses of each successively greater even-numbered HD (Fig. 1b, dashed line). Nonspecific amplifications were minimized beyond mismatches of ~6 HD and greater. Indeed, the likelihood of amplification was expected to be lower than the likelihood of hybridization since in wet experimental conditions a primer samples the reaction volume with the potential to interact with other strand regions, and also must interact with a DNA polymerase molecule to carry out the amplification.

Systems based upon such stringent criteria have had success with small-scale systems, but this criterion constrains the set of potential non-interacting addresses to a few thousand from the theoretical maximum of $4^{19}$ (for 20-nt addresses)[12–14,16] (Fig. 1b, inset). This severely limits the functionality of DNA storage systems. We hypothesized that rather than viewing nonspecific interactions as a hindrance, they could instead be potentially useful if controllable. In particular, it could be possible to tune the access of different subsets of DNA strands by simply changing environmental conditions while using the same file-access primer.

Toward this goal, we considered how biomolecular interactions are governed by thermodynamics (Fig. 1c), with more negative Gibbs free energy ($\Delta G$), lower temperature, or higher primer concentration leading to more template binding sites being bound. Sequences with a higher HD have a less negative $\Delta G$ (they are less favorable to bind thermodynamically) but this can be compensated by changes in temperature or primer concentration. Embedded in this equilibrium equation also is that the

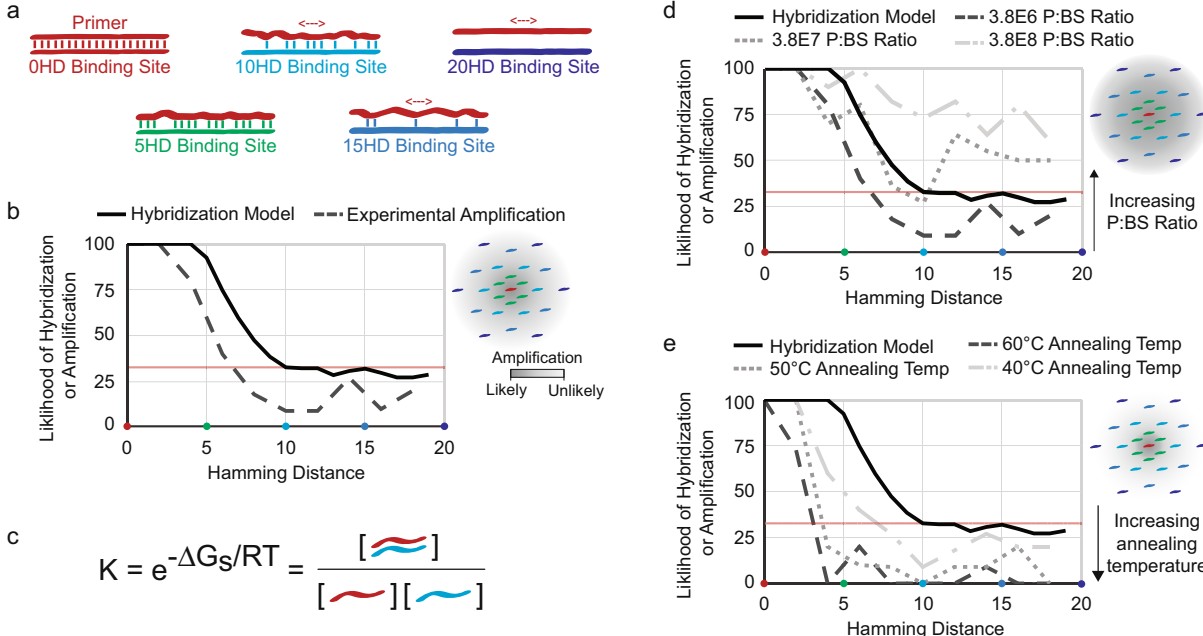

**Fig. 1 Stringency of PCR reactions is tunable via annealing temperature and primer concentration. a** File address sequence similarity is inversely proportional to Hamming distance (HD—total number of nucleotide differences in a given sequence). While perfectly matching (0 HD) primers tightly bind their complementary binding sequence, primers with increasing HDs can still bind with a gradually diminishing effect. **b** A thermodynamic model shows that the likelihood of hybridization (black trace) reaches a plateau (red line) around 10 HD and remains level out to 20 HD. Likelihood of amplification (gray dashed line) is represented as a percent of the ten sequences experimentally tested at each HD (0, 2, 4, 6, 8, 10, 12, 14, 16, 18, 20) that created PCR product. **b**, **d**, **e**, spherical insets: Visualization of address space with a perfect primer-address match at the center in red. DNA storage systems currently implement addresses that are at least 10 HD (light blue strands) apart. This disregards and wastes much of the potential address space (i.e., green strands). Gray shading indicates the likelihood of hybridization/amplification with a red primer. **c** The equilibrium constant of thermodynamically controlled interactions can be computed based on the sequences' Gibbs Free energy (ΔG), the gas constant (R), and the PCR annealing temperature (T). Annealing temperature and primer concentration impact the amount of template amplified via PCR. **d** Primer concentration, represented by the primer to binding site (P:BS) ratio, is experimentally varied at a constant 50 °C annealing temperature. The likelihood of nonspecific amplification increases with increasing P:BS ratio. **e** The annealing temperature is experimentally varied at a constant 1.9E7 P:BS ratio. The likelihood of nonspecific amplification decreases with increasing annealing temperature. Source data are provided as a Source Data file.

equilibrium constant itself can be dependent on other environmental factors such as ionic strength and the presence of detergents. Based on thermodynamics[24–26] and significant practical work in molecular biology and biochemistry[27–29], we tested how a range of temperatures and primer concentrations would shift the likelihood of PCR amplification. As expected, lower annealing temperatures and higher primer concentrations increased nonspecific amplifications, while higher annealing temperatures and lower primer concentrations decreased nonspecific amplifications (Fig. 1d, e and Supplementary Table 1).

**Thermodynamics tune amplification within competitive PCRs.** DNA strands in a storage system do not function in isolation, so we designed a competitive system with two unique template strands that had closely related addresses. In this reaction, we added a single 20-nt PCR primer pair used to amplify both strands: a 200 bp template with perfectly complementary primer binding sites (0 HD) and a 60 bp template with primer binding sites containing two mismatches (2 HD). In this competitive context using only one pair of PCR primers, only the 0 HD strands were amplified using stringent conditions (e.g., high annealing temperature and/or low primer concentration). Both 0 HD and 2 HD strands were amplified using promiscuous conditions (e.g., low annealing temperature and/or high primer concentration) (Fig. 2a).

To further tune the relative yield of 0 and 2 HD strands, strands with six distinct 2 HD forward primer binding addresses and five distinct 2 HD reverse primer binding

addresses, paired in all combinations, were amplified with the same primer set in PCRs at both stringent and promiscuous conditions. This yielded a range of ratios of promiscuous to stringent amplifications (Fig. 2b). Interestingly, the 2 HD addresses that exhibited tunability when varying annealing temperature also tended to be more likely to exhibit tunability when varying primer concentration.

**Implementing File Preview of jpeg images through thermodynamic swings.** We hypothesized that this tunable promiscuity could provide a useful framework for organizing data and implementing functionalities. We focused on engineering a practical data access function, File Preview (Fig. 3a). For an image, this could be implemented where stringent PCR conditions would amplify and access only a subset of strands encoding a low-resolution pixelated Preview version (or thumbnail) of an image. In contrast, promiscuous PCR conditions would amplify both the Preview strands and the rest of the strands comprising the full image. The same exact primers would be used in both stringent and promiscuous conditions. We asked if this tunability could be applied to entire files (NCSU Wuflab logo—25.6 kB, two Wright glider images—27.9 and 30.9 kB—Fig. 3f left and right, respectively, Earth—27.2 kB) rather than just toy DNA strands. Furthermore, we expanded our screen for primers and addresses and asked if this principle of tunable promiscuity could be extended to more distant HDs to create multiple Preview layers. We screened four 20-nt primer pairs and up to 30 distinct 0, 2, 3, 4, and 6 HD addresses per pair. We screened them individually and in

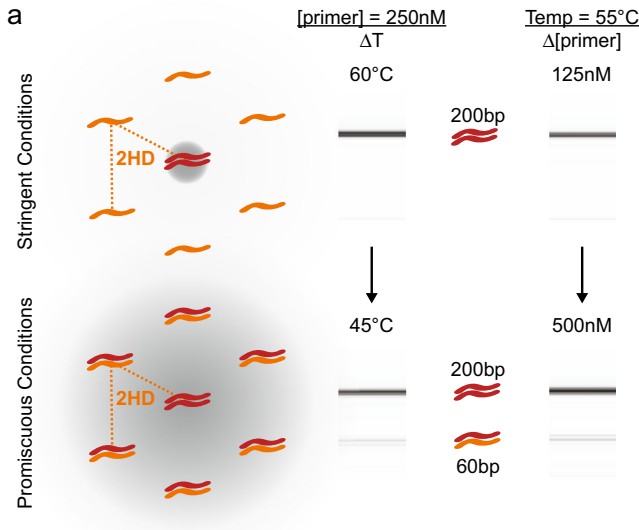

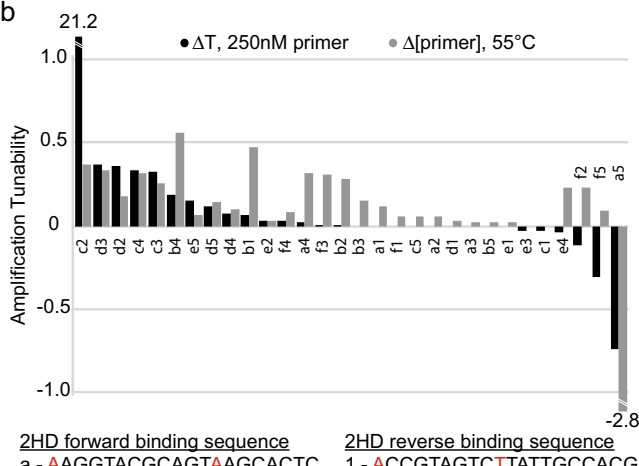

2HD forward binding sequence
a - AAGGTACGCAGTAAGCACTC
b - CAGGGGACACAGTTAGCACTC
c - CAGGTACACAGTGAGCACTC
d - CAGGTACGCAGTTGCAATC
e - CAGGTCCGCAGTTAGAACTC
f - CATGTTCGCAGTTAGCACTC

2HD reverse binding sequence
1 - ACCGTAGTCTTATTGCCACG
2 - TCCGTAGTCAGATTGCCATG
3 - TGCGTAGTCATATAGCCACG
4 - TTCGTAGTCATATGGCCACG
5 - TTCGTAGTCGTATTGCCACG

**Fig. 2 Thermodynamic tuning of amplification within competitive PCRs.**
**a** Strands with 2 HD binding sites (60 bp, orange) are screened for nonspecific amplification in a competitive reaction against 0 HD strands (200 bp, red). Two stringent conditions are individually tested (top row): (left) 250 nM primer at 60 °C and (right) 125 nM at 55 °C. The promiscuous conditions individually tested (bottom row) are (left) 250 nM primer at 45 °C and (right) 500 nM at 55 °C. Gray spheres encompass strands that are expected to be amplified, and the gel electrophoresis lanes show experimental results. **b** A screen of a library of sequences is conducted to find sequences to be used in scaling to full files and databases. Each forward binding sequence (letters a–f) is paired with every reverse binding sequence (numbers 1–5) in the reactions described in **a**. Amplification tunability is defined as the difference in the ratio of mismatch (60 bp) strands to perfect match (200 bp) strands from promiscuous to stringent conditions. Positive values represent tunability in the expected direction. Tunability using annealing temperature (black) and primer concentration (gray) are shown. Source data are provided as a Source Data file.

competitive reactions using a diverse range of PCR conditions incorporating salt concentrations, detergents, temperature, primer concentration, number of unique mismatch strands present, and size and ratios of template strands (Supplementary Tables 1 and 2 and Supplementary Figs. 1 and 2).

Based upon these results, we designed files containing 0, 4, and 6 HD strands (Supplementary Fig. 1). Selecting the most consistent primer and its variable addresses, the Preview data strands were encoded with the fully complementary primer binding addresses (0 HD) while the rest of the file was encoded with the 4 HD (Intermediate Preview) and 6 HD (Full Access) addresses. The most stringent condition successfully accessed only the Preview image (Fig. 3b). Furthermore, the distribution of sequencing reads showed this Preview condition cleanly accessed only the Preview strands (Fig. 3c, top). When conditions were made less stringent, both 0 and 4 HD strands were accessed as expected, and the intermediate preview image with the higher resolution was obtained. However, when we attempted to access the full file, we did not obtain any 6 HD strands. Instead, we discovered that there were problematic sequences in the data payload region that had been inadvertently encoded to be only 5 HD from the primer sequence. While the full file was therefore not accessed, this accident serendipitously revealed that a relatively sharp transition of just 1 HD (between 4 and 5 HD) could be cleanly achieved between the intermediate and full file access conditions (Fig. 3c). We also found in this experiment that because the 0 HD strands amplified efficiently in all conditions, it often dominated the distribution of sequencing reads. We, therefore, found that increasing the physical copy number of mismatched strands (alternatively one could encode more data in the higher HD partition of the file) resulted in a more even sequencing read distribution between 0 and 4 HD strands. Furthermore, by using more promiscuous access conditions, the balance of 0 and 4 HD strands that were accessed could be tuned and evened out (Fig. 3c, middle vs. bottom).

To explore these transitions and develop more informed control over them, diverse factors were individually varied to determine their impact on reaction specificity/promiscuity (Fig. 3d, e and Supplementary Figs. 2 and 3). 0, 2, 4, and 6 HD strands were used, each having unique restriction sites that allowed for digestion and facile quantification of each strand type by capillary electrophoresis. The accidental 5 HD strands were still present so their contributions were quantified as well. The most important factor in Preview tunability was PCR annealing temperature, with a low temperature (40–45 °C) resulting in an increased proportion of mismatched strands when compared to high annealing temperatures (55–60 °C). Other parameters and reagents were nonetheless important for fine-tuning the system. Primer and magnesium chloride ($MgCl_2$) concentrations had inverse relationships with specificity, while potassium chloride (KCl) concentration exhibited a direct relationship to specificity up to 150 mM (beyond which PCR amplification was completely inhibited, Supplementary Fig. 3a, b). In aggregate, a gradient of distinct conditions were identified that were able to specifically access 0, 0–2, 0–2–4, and 0–2–4–5 HD strands as well as successfully decode low, intermediate, and higher-resolution images (Fig. 3f, g and Supplementary Fig. 3c).

In a true data storage system, each file will be a small fraction of the total data. Biochemical interactions may be affected by the high background of other DNA strands and potential nonspecific interactions; we, therefore, asked if File Preview could still function in a high background system. A text file encoding the United States Declaration of Independence was amplified via error-prone PCR[30] to create strands equivalent to 1.5 GB of data (Supplementary Fig. 4a), and each image file (NCSU Wolf, two Wright glider images, Earth) was amplified in the presence of this nonspecific, noisy background (Fig. 3h). In this setting, the Preview strands (0 HD) were merely ~0.036% of the total number of strands present in the reaction. Encouragingly, we were still able to reliably amplify and decode the Preview strands for each of the four files using stringent PCR access conditions. When promiscuous PCR

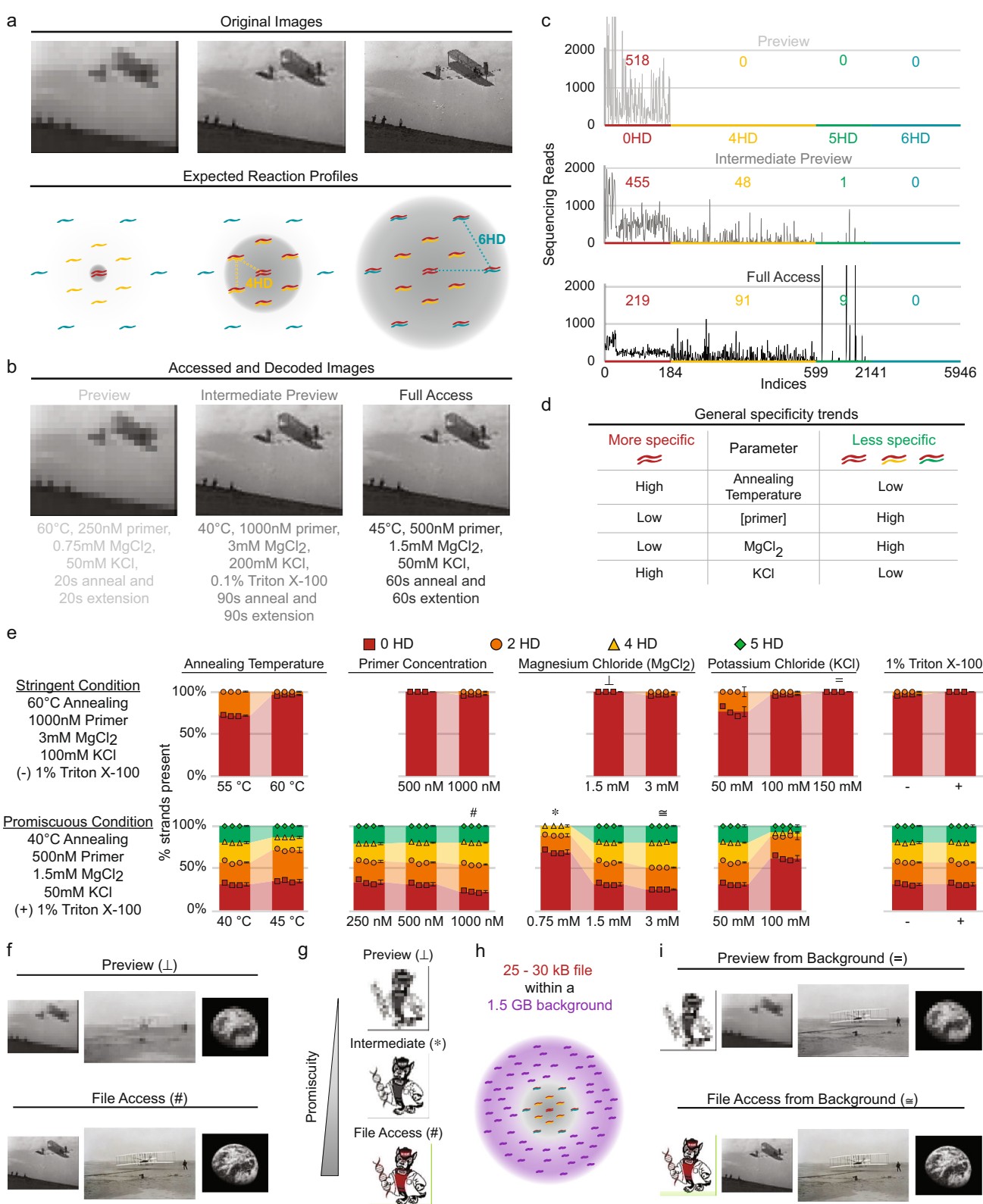

conditions were used all four files were able to be accessed, decoded, and displayed without background contamination (Fig. 3i and Supplementary Fig. 4b).

**File preview can reduce next-generation sequencing costs**. This system provides an innovative functionality for DNA data storage systems; however, it is important to consider what the

potential benefits and tradeoffs of this system may be from a practical and quantitative perspective. When implementing Preview there are two main tradeoffs to consider: physical storage density and sequencing cost (Fig. 4). In our current balance of Preview versus full-access strands, we are previewing ~5% of a file's strands (5% File Preview). This requires 100× more copies of each unique 4 HD strand than each unique

**Fig. 3 Implementing File Preview of jpeg images through thermodynamic swings. a** Subsets of strands encoding increasingly more data to create a higher-resolution image. The expected reaction profiles show that higher HD strands (0 + 4 and 0 + 4 + 6 HD, middle and right, respectively) must be amplified to obtain the desired resolution image. **b** Experimental results showing high stringency to low stringency conditions are used to access and decode images. Intermediate Preview and Full Access result in identical images due to nonspecific 5 HD binding sites interfering with amplification of 6 HD strands. **c** Next-generation sequencing read counts versus strand index number for preview, intermediate preview, and full-access conditions. The average read depth per strand index is listed above the corresponding HD regions. Most 5 HD indices appear within 6 HD strands, but their truncated amplification products are uniquely distinguished by NGS. Including 5 HD products overrepresents the number of unique file sequences: 5946 indices represent the number of amplification products; only 4405 unique strands actually encode the file. **d** A screen of environmental parameters reveals trends controlling the specificity of PCR amplification (data shown in Supplementary Fig. 3a, b). **e** Environmental parameters are independently varied from one stringent and one promiscuous base condition. The percentage of 0 (red), 2 (orange), 4 (yellow), and 5 HD (green) strands are measured by capillary electrophoresis. Wuflab logo file data shown here. Data points represent triplicate reactions. Error bars represent standard deviation. The Center of error bars represents the mean of the triplicate reactions. Source data are provided as a Source Data file. **f** Preview (⊥) and File Access (#) conditions from (**e**) are selected to access three jpeg files, followed by NGS analysis. All files are successfully decoded. The image resolutions all increase from the Preview to the File Access conditions. **g** A gradient of Preview conditions is also achieved. Preview (⊥), Intermediate (*), and File Access (#) conditions from (**e**) successfully accessed the Wuflab logo as measured by NGS analysis. **h** A text file containing the Declaration of Independence is amplified using error-prone PCR to create a noisy, nonspecific background equivalent to 1.5 GB of data. **i** Preview works in the context of the GB-scale background, and all files are successfully decoded after amplification from the 1.5 GB noisy background and NGS.

0 HD strand (1:100 ratio) to account for differences in PCR efficiency. With this current configuration, a file in which 5% of the strands are used for Preview requires 95× of the physical space to be stored (Fig. 4a, black line) compared to normal encoding. Further reducing the copy number of full file strands by a factor of ten (1:10 ratio) or twenty (1:5 ratio) allows a file to be stored in 9.5× or 4.8× of the theoretical minimum physical space, respectively (Fig. 4a, gray dashed and light-gray dashed lines). This loss of physical efficiency is tunable based on the percent of the file to be Previewed and, subsequently, the number of excess copies of each unique full file strand to be stored. For example, when the Preview strands account for a smaller fraction of a file (~0.1–1%), the total number of full file strands will already be in a sufficient ratio to Preview strands to account for PCR efficiency differences; therefore, excess copies will not need to be stored. This removes the negative tradeoff in-storage density. In the future, for any desired percentage of a file that one wishes to encode with Preview strands, one may be able to match access conditions, polymerase type, or primer selection so that all unique strands are present at equivalent copy numbers.

With regards to cost, when searching for a file in a database or recovering only key portions of data in a series of files, costs may be lowered by requiring the sequencing of fewer strands when quickly Previewing a file (or multiple files) rather than needing to sequence entire files. To understand this tradeoff, envision a small database with 15 very similar files where: the full contents of the files are unknown, all 15 pairs of access primers are known, and a user is trying to find and sequence a target file of interest from amongst these 15 files based upon information that is not included in any metadata system. Without File Preview, one would potentially sequence 15 full files before finding the correct one. Using File Preview, one would sequence only the Preview strands of each of the 15 files until the correct file was found. Then that full file would be sequenced. Assuming all 15 files were searched, it would cost 85.3% less to find and fully sequence a file using a 5% Preview system (5% of all unique strands are Preview strands) compared to a normal encoding system (Fig. 4b). This cost advantage only increases as the percentage of strands encoding the Preview strands decreases, and as the number of files needed to be searched increases. Encouragingly, even without further engineering the access conditions, by reducing the percent of the file being Previewed from 5 to 1% it would cost 91.7% less to find and fully sequence a file from the 15-file library using the Preview system compared to a normal encoding system.

## Discussion

The File Preview function is practical in that it reduces the number of strands that need to be sequenced when searching for the desired file. This will reduce the latency and cost of DNA sequencing and decoding. Consequently, one will be able to search a database of files much more rapidly and cost-effectively using Preview than if each file needed to be fully sequenced. Beyond the Preview function, this inducible promiscuity technology could be used for many other data or computing applications. It may have broad application to how data is managed or organized in a file system. For example, files could be differentially encoded to make it cheaper and easier to access frequently versus infrequently used data. Another interesting use case is support for deduplication of data, a ubiquitous need in large and small data sets in which replicated blocks of data are detected and optimized[31]. Rather than storing many copies of duplicated data, a single copy could be shared amongst files by taking advantage of the promiscuous binding.

Although we initially designed our File Preview system to include 0, 2, 4, and 6 HD file addresses for each file, there were problematic sequences that arose within the data payload region. Specifically, when two particular codewords were adjacent to each other their sequences combined to create a binding site 5 HD from one of the accessing primers. While this was unintended, similar sequences can be avoided in the encoding process using thorough quality control measures that screen through all possible codeword combinations. Primer sequences are typically designed to be >8 HD from data payloads;[16] accordingly, we expect data encoding densities can remain unchanged when implementing File Preview since only a single primer pair is used per file.

However, it is important to note and consider that using more promiscuous conditions could increase off-target interactions more generally in the data payload regions even if all <10 HD sequences are avoided. This possibility should be investigated in the future as part of expanding our overall understanding of off-target interactions, particularly in extreme-scale systems. However, our work (Figs. 2 and 3) suggests that the presence of <10 HD addresses in File Preview systems will outcompete interactions with higher HD off-target sequences that may be present in data payload regions. For example, while 4, 5, and 6 HD binding sites were very similar in sequence, stepwise decreases in accessing each HD set could be cleanly achieved by tuning PCR conditions. Thus, the chances of off-target interactions are most likely to occur within strands of the same file that have higher HD addresses rather than in strands of an undesired file. In addition,

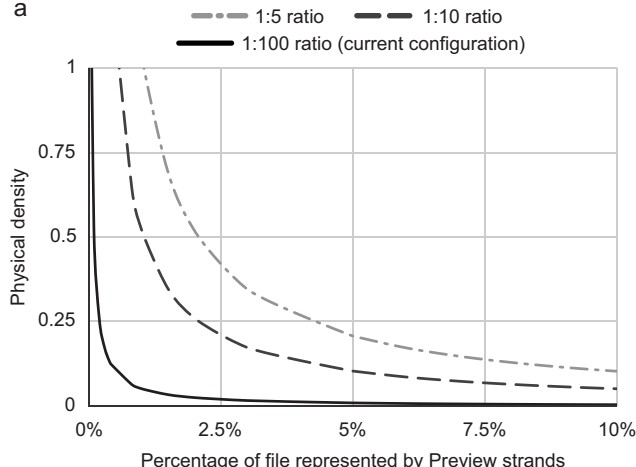

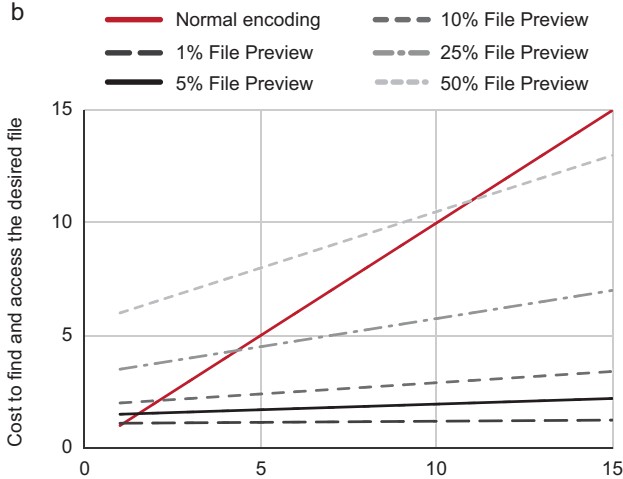

**Fig. 4 File Preview can be implemented at similar densities while reducing sequencing costs. a** Physical density to store a given file using Preview encoding normalized to the physical density of normal encoding. When holding the ratio of total preview strands to total full file strands constant at 100 (black line, current configuration), 10 (gray dashed line), or 5 copies (light-gray dashed-dot line), the physical density exponentially decreases as more of the file is stored in the Preview strands. **b** File Preview can cost-effectively find a file in a library. Cost to find a file is defined as the normalized cost to fully sequence an entire file within a library, e.g., sequencing a 15-file database costs 15 on the y axis. File preview can be used to quickly and cheaply find a file by sequencing a fewer total number of strands than is needed in a normally encoded library. Decreasing the percentage of each file stored in Preview strands further decreases the cost of finding a file. Source data are provided as a Source Data file.

we did not observe off-target access from the randomized 1.5 GB of data background in Fig. 3h, i. Despite this, it would be prudent in the future to carefully assess within extreme-scale systems how increasing promiscuity of access conditions statistically increases the chances of inadvertently accessing strands from off-target files.

While previous DNA-based storage systems draw inspiration from conventional storage media and have had success, shifting the design paradigms to naturally leverage the intrinsic structural and biophysical properties of DNA holds the significant promise that could transform the functionality, practicality, and economics of DNA storage. This work provides an archetype for a biochemically driven and enhanced data storage system.

## Methods

**Hybridization model.** Hamming distance is frequently used as a metric in the design of primers to support random access of files in DNA-based storage systems because a high Hamming distance is an effective proxy for low hybridization likelihood. Hamming distance is a measure of how many symbols differ between two symbolic strings and in our case, the strings of interest are two DNA primer sequences.

When analyzing two primers, $p_1$ and $p_2$, we compared their Hamming distance directly by lining up the sequences and counting the positions in which they are different. Hence, a Hamming distance of 0 means that the two primers were in fact the same sequence. If the Hamming distance was equal to the length of the primer, then every position was different. However, in terms of hybridization, we were interested in whether $p_1$ will bind to the reverse complement of $p_2$, as that binding site was present on the data strand. For convenience, we describe the Hamming distance of the two coding primers, but for hybridization, we analyzed the hybridization likelihood for $p_1$ against the reverse complement of $p_2$. Hence, a Hamming distance of 0 implies hybridization was guaranteed, however, a high Hamming distance implies that hybridization was unlikely, although caveats existed. For example, if $p_1$ was the same as $p_2$ but merely shifted over one position, it had a high Hamming distance but a low edit distance. Such a high Hamming distant primer almost certainly bound due to low edit distance. To ensure that low edit distances do not skew the findings, primers with a much lower edit distance than Hamming distance were screened.

While high Hamming distances of 10 or more were common in past literature, low Hamming distances and their relationship to hybridization were of particular interest to our design. To better understand the potential of exploiting primer binding among similar but non-complementary primers, an in silico analysis was used to predict the likelihood of primer hybridization as a function of Hamming distance. Our approach was based on a Monte Carlo simulation that considered the likelihood of hybridization among many primer pairs. One primer was designed specifically for data storage using procedures common in the field, namely it must have had GC-balance between 40 and 60%, melting temperature between 50 and 55 °C, and avoided long homopolymers. Then, it was randomly mutated into a new primer with varying Hamming distances, from 1 to $N$, where $N$ was the length of the string. The mutated primer was produced by generating a random index from 1 to $N$ with equal likelihood and randomly picking a new base for that position from the other three bases with equal probability. The mutation process was repeated until a primer with a suitable distance was achieved. Primers with a much lower edit distance were screened in this step, and it is worth noting that such primers had a very low probability due to the probabilistic nature of the mutate step; only a handful was observed over all trials. Using NUPACK's complex tool, the ΔG for the complex arising from the original primer binding to the reverse complement of the mutated primer was estimated[23]. Negative values beyond a threshold of −10 kcal/mol were interpreted as binding in our analysis. The Monte Carlo simulation included at least 10,000 trials for each given Hamming distance to estimate the likelihood of hybridization. The percentage of mutated primers with a high chance of hybridizing for each Hamming distance is reported as the hybridization % in Fig. 1b.

The python program that performed this analysis is included in our code repository as part of the Supplementary Material[32].

**Hamming distance primer design.** Primers were selected for use in File Preview using a similar screening process as that for the Hybridization Model. However, instead of generating many trials, only a handful of primers were produced at each desired Hamming distance. These primers were then subjected to additional experimental screening.

**Experimental model verification—qPCR amplification.** Using one primer sequence as the 0 Hamming distance amplifying primer, 10 variable strand addresses at each even-numbered Hamming distance were used as template strands for qPCR amplification (Supplementary Table 1). All strands were amplified using the same primer pair since they contained the same forward primer binding sequence while varying the reverse primer binding sequence. Reactions were performed in 6 μL format using SsoAdvanced Universal SYBR Green Supermix (BioRad). A range of primer concentrations (125–500 nM), template strand concentrations (2E3-2E6 strands/μL), and annealing temperatures (40–60 °C) were tested. Thermocycler protocols were as follows: 95 °C for 2 min and then 50 cycles of: 95 °C for 10 s, 40–60 °C for 20 s, and 60 °C for 20 s followed by a melt curve increasing from 60 °C to 95 °C in 0.5 °C increments held for 5 s each. Data were analyzed using BioRad CFX Maestro. Cq value (i.e., cycle number at which a sample reached a defined fluorescence) and melt curve data (i.e., temperature a sample was denatured while being heated) were used for analysis. Successful amplifications were defined as crossing the Cq threshold before the negative control while also creating an individual peak (i.e., single product) on the melt curve.

**Competitive PCR primer reactions.** Using four distinct primer pairs as the 0 Hamming distance amplifying primers, 5–30 unique strands (60 bp) containing variable address pairs at 2, 3, 4, or 6 Hamming distance were tested as template

strands alongside 0 HD strands (200 bp) in competitive qPCR amplifications (Supplementary Table 2). All strands designed using the same original primers were amplified using the 0 HD primer pair. Reactions were performed in 6 μL format using SsoAdvanced Universal SYBR Green Supermix (BioRad). Template strand concentrations were in equal copy number concentration for the 0 HD and variable HD strands (1.67E5 strands/μL). A range of primer concentrations (125–500 nM) and annealing temperatures (40–60 °C) were tested. Thermocycler protocols were as follows: 95 °C for 2 min and then 50 cycles of: 95 °C for 10 s, 40–60 °C for 20 s, and 60 °C for 20 s. Final products were diluted 1:60 in 1×TE before analysis using high-sensitivity DNA fragment electrophoresis (Agilent DNF-474; Advanced Analytical 5200 Fragment Analyzer System; Data analysis using Prosize 3.0). The ability of a primer to variably amplify a strand with a nonspecific primer binding site at different PCR conditions, or amplification tunability, was calculated using the following equation (concentrations in nmole/L):

$$\text{Amplification tunability} = \triangle \left( \frac{[\text{nonspecific strand}]}{[\text{specific strand}]} \right) = \left( \frac{[\text{ns strand}]}{[\text{s strand}]} \right)_{\text{Promiscuous}} - \left( \frac{[\text{ns strand}]}{[\text{s strand}]} \right)_{\text{Stringent}}$$

(1)

**JPEG encoding for File Preview operations**. File Preview was performed on JPEG images due to their widespread popularity, their small storage footprint, and their support for organizing data within a file that works synergistically with the goals of File Preview in this work. In particular, JPEG's progressive encoding[33] allowed for image data to be partitioned into scans by the color band and by frequency. Through careful organization of the file, a low-resolution grayscale image was constructed from a small percentage of the file's data, or an increasingly higher-resolution image was obtained from reading a greater percentage of the file[32]. For the File Preview operations, the JPEG information was arranged in such a way that a 0 HD access pulled out a small amount of data and produced a low-resolution image. By tuning the access conditions as described, more of the file was accessed and a greater resolution image was produced.

The most important aspects of the JPEG format are described for the sake of explaining how Preview works. JPEG holds information in three color bands known as Y, Cb, Cr that together encode information for all pixels in an image. Y represents luminosity, Cb is a blue color band, and Cr is a red color band. Together, these components may represent any conventional RGB color. Each pixel of an image can be thought of as having a tuple of Y, Cb, and Cr components although they are not actually stored that way.

JPEG does not store images in a naive matrix of (Y, Cb, Cr) pixel values. This would waste storage since many pixels have the same color. Instead, each 8 × 8 block of pixels from each color band are converted into a frequency domain representation using the 2-D Discrete Cosine Transform (DCT). The 2-D DCT has the interesting effect of partitioning the data into low-frequency and high-frequency components. Each 8 × 8 block becomes a linearized vector of 64 values ordered from low frequency to high frequency. The first value in the vector is known as the DC value because it represents an average value across the original 8 × 8 pixel block. For example, if the original 8 × 8 blocks were entirely white, the Y band would have a DC value of 255, indicating the average value over the block was white. The remaining 63 entries represent the higher frequency components known as the AC band. For an all-white block, the rest of the vector would be 0, indicating no other content.

In a progressive encoding, each color band is encoded in scans. A scan is the aggregation of all values from a given position in the linearized vector across all 8 × 8 blocks. For example, the first scan of a file would include all of the DC values from the Y band across all 8 × 8 blocks. The scan of DC values for a given band is given as Y[0], Cr[0], or Cb[0]. The Y[0] scan by itself is essentially a low-resolution grayscale image. Cr[0] and Cb[0] would add low-resolution color information.

The DC scans precede the AC scans. The AC scans group the following AC components together, and these scans could include a single value from the linearized vector or multiple values. For example, Y[1:5] would include indices 1 through 5 of the linearized vectors taken from all 8 × 8 blocks in the Y band. All indices from 1 through 63 must be included in at least one scan. This is repeated for all bands. The JPEG standard additionally compresses each scan to save storage space, but the details of that mechanism are not pertinent to Preview and are omitted. Furthermore, the scans follow the standard and are stored in compressed form.

The JPEG files were first encoded into 42 scans: Y[0], Cr[0], Cb[0], Y[1:5], Cb[0],Cr[0], Y[6:10], Y[11:15], Y[16:20], Y[21:25], Y[26:30], Y[31:35], Y[36:40], Y[41:45], Y[46:50], Y[51:55], Y[56:60], Y[61:63], Cb[1:5], Cr[1:5], Cb[6:10], Cr[6:10], Cb[11:15], Cr[11:15], Cb[16:20], Cr[16:20], Cb[21:25], Cr[21:25], Cb[26:30], Cr[26:30], Cb[31:35], Cr[31:35], Cb[36:40], Cr[36:40],Cb[41:45], Cr[41:45], Cb[46:50], Cr[46:50], Cb[51:55], Cr[51:55], Cb[56:60], Cr[56:60], Cb[61:63], Cr[61:63].

The scans were grouped into partitions. Wuflab logo and Wright Glider 2 had four partitions, and Wright Glider 1 and Earth had three partitions. In all cases, the first and second partitions, if accessed alone, provided low-resolution images that are recognizable as the image. For the Wuflab logo and Wright Glider 2 files, the third partition contained all remaining scans. For the others, the third partition added DC color information and some higher frequencies of the Y band to improve image quality, and the fourth partition contained all remaining scans.

Each partition was treated as a block of data and encoded into DNA as a unit. Each partition was tagged with primers. Higher numbered partitions were given primers with a greater Hamming distance.

The encoding process is described in Supplementary Fig. 7. Each partition was encoded into DNA using a multilevel approach. First, the JPEG file was partitioned into scans. Then, each partition was divided into blocks of 1665 bytes, which were interpreted as a matrix with 185 rows and 9 columns with one byte per position. Blocks smaller than 1665 bytes at the end of a file or partition were padded out with zeros. An RS outer code with parameters of [$n = 255$, $k = 185$, $d = 71$] added additional rows to each block to compensate for the possible loss of strands within a block. Each row was given a unique index that was two bytes long. Then, each row was appended with error correction symbols using an RS inner code given as [$n = 14$, $k = 11$, $d = 4$] that protected both the data and index bytes.

Each row of a byte was converted into a DNA sequence using a comma-free code that mapped each byte to a unique codeword sequence. The codewords were designed using a greedy algorithm to be GC-balanced and have an edit distance of at least two to all other codewords. Each codeword had a length of 8 nts. The last step was the appending of primers to each end of the sequence and insertion of a restriction enzyme cut site in the middle of the strand. Each partition of the JPEG file used different primer binding sites, so these primer sequences were given as inputs for each partition as it was encoded.

An additional set of flanking primers were added to each strand to enable the entire library to be amplified at once using a single common primer. The final set of strands for each file were synthesized into a DNA library.

**PCR condition screening and File Preview**. The four-file synthetic DNA library was ordered from Twist Biosciences. Flanking primer amplifications unique to each subset of strands (Supplementary Table 3) were optimized and the resulting products were used in screening and preview reactions. Each subset of strands within a file encodes an increasing percentage of the stored image and contains a unique restriction enzyme cut site to allow for rapid sample analysis. It was determined that each block of data encoded in strands with increasing Hamming distance binding sites (2, 4, and 6 HD), needed to be physically stored with extra copies of the nonspecific strands: 10×, 100×, and 1000×, respectively. A screen of variable PCR conditions was conducted on files from the library prior to preview and full-access reactions. Reactions were performed in 6–50 μL format using SsoAdvanced Universal SYBR Green Supermix (BioRad) or Taq polymerase (Invitrogen). Conditions varied during testing include: annealing temperature (40–60 °C), annealing and extension timing (20–90 s), number of cycles (25–40), primer concentration (62.5–1000 nM), polymerase concentration (0.5–2× recommended units), dNTP concentration (200–800 μM), MgCl₂ concentration (0.75–3 mM), KCl concentration (50–200 mM), and absence or presence of 0.1–1% Triton X−100, 0.1–1% BSA, 0.1–1% Tween−20, 2–8% DMSO, 0.1–3.5 mM Betaine, or 2% DMSO plus 0.1 mM Betaine. Reaction products (1 μL) were added to restriction enzyme reactions to cut 0, 2, 4, or 6 HD sections of the products. Digestion products were diluted 1:3 in 1×TE for analysis using high-sensitivity DNA fragment electrophoresis (Agilent DNF−474; Advanced Analytical 5200 Fragment Analyzer System; Data Analysis using Prosize 3.0). Quantification data were taken directly from Fragment Analyzer. Undigested preview, full access, and intermediate samples were then analyzed via Illumina Next-Generation Sequencing (Genewiz and AmpliconEZ).

**Error-prone PCR**. Template DNA was amplified using 0.5 μL of Taq DNA polymerase (5 units/μL, Invitrogen) in a 50 μL reaction containing 1× Taq polymerase Rxn Buffer (Invitrogen), 2 mM MgCl₂ (Invitrogen), the sense and antisense primers at 1E13 strands each, and dATP, dCTP, dGTP, dTTP (NEB), dPTP (TriLink), and 8-oxo-dGTP (TriLink), each at 400 mM. PCR conditions were 95 °C for 30 s, 50 °C for 30 s, and 72 °C for 30 s for 35 cycles with a final 72 °C extension step for 30 s.

**Calculation of data quantity of error-prone background**. In Fig. 3h, i and Supplementary Fig. 4, we refer to background size (GB). For clarity and ease of comparison, this value was calculated based on the total number of DNA strands. Each strand is comprised of 200 nts, 20 of which are used for each primer sequence, 16 for the index, and 8 for the checksum. Eight nts comprise each 1-byte codeword. Thus, each strand addressed with a single primer pair contains 17 bytes of data. We assumed a 10-copy physical redundancy per unique strand to provide a conservative estimate for a realistic system where multiple copies of each strand would likely be needed to avoid strand losses and inhomogeneous strand distributions. Thus, the total background size is divided by 10.

**Next-generation sequencing and File Preview decoding**. FASTQ files obtained from sequencing were all decoded successfully into images. Decoding occurred in the reverse order shown in Supplementary Fig 7. Files were reconstructed by placing all data blocks and JPEG file partitions into the correct order based on their index. Since error correction was applied separately to each partition, each partition succeeded or failed at partition boundaries. If a partition was incomplete, it was omitted from the JPEG image. As long as omitted partitions were the latter partitions taken from AC scans, their absence only reduced the quality of the JPEG image and made it appear lower resolution or grayscale, depending on the scans

that were lost in the partition. However, if the first partition in the file was missing or too erroneous to decode, the image would be unreadable. No experiment yielded an undecodable or unreadable image. The successfully decoded images are shown in Fig. 3b, f, and h.

To gain deeper insight into which strands were sequenced and their relative abundance, a clustering analysis was performed on all sequenced reads[32]. The Starcode algorithm is an open-source and efficient algorithm for performing an all-pairs search on a set of sequencing data to find all sequences that are within a given Levenshtein distance to each other[34]. To derive the number of reads for each encoded strand in the library, the algorithm was seeded with 20 copies of each strand from the library. The Starcode algorithm was additionally given the following parameters: Levenshtein distance set to 8 edits, the clustering algorithm set to message passing, and the cluster ratio set to 5. The Levenshtein distance parameter defines the maximum edit distance allowed when determining whether a strand belongs to a cluster. The clustering algorithm attributed all reads for a given strand $S$ to another strand $V$ provided that the ratio of $V$'s reads to $S$'s reads were at least the cluster ratio. Hence, providing 20 copies of each expected strand ensured that each was well represented during clustering such that it was considered a centroid. With the clusters formed, each centroid was interrogated to make sure that it was a strand from the library and not an unexpected strand present during sequencing. If the centroid matched an expected strand defined by the encoded file (s), the number of reads for that strand was adjusted to match the size of the cluster less than 20. These results are reported in Fig. 3c.

**Reporting summary**. Further information on research design is available in the Nature Research Reporting Summary linked to this article.

## Data availability

Supplementary Information are available with this paper and at https://github.com/dna-storage/ncomm-file-preview/releases/tag/v0.1-alpha along with the next-generation sequencing data for Figs. 3b, 3c, 3f, 3g, 3h, and Supplementary Figs. 3c and 4b. Any other data are available upon reasonable request. Source data are provided with this paper.

## Code availability

The software algorithms we developed to perform the reported analyses for Figs. 3b, 3c, 3f, 3g, and 3i are available at https://doi.org/10.5281/zenodo.4747693 and https://github.com/dna-storage/ncomm-file-preview under a permissive open-source license with instructions for installation. We implemented code in python using many standard open-source packages tested for compatibility with python versions 3.6 to 3.9. The dependences are documented in the form of a python requirements.txt file that guides the installation of additional dependent software packages. NUPACK 3.0 was used to develop the primer hybridization model. We used python-Levenshtein 0.12.0 for edit distance calculations. The open-source sequencing data clustering software Starcode Algorithm was used to aid in the process of determining read counts for strands. The version used in the analysis is the master branch that can be accessed at https://github.com/gui11aume/starcode. A docker file is available to make setup on a wide variety of systems easier.

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

## Acknowledgements

We thank Kevin N. Lin, Karishma Matange, Magdelene Lee, and Doug Townsend for fruitful discussions during group meetings. We thank the NC State Biotechnology Program for providing the Wuflab logo. We also thank Joshua Emrick for his helpful comments on the manuscript. This work was supported by the National Science Foundation (Grants CNS−1650148, CNS−1901324, ECCS 2027655) and a North Carolina State University Research and Innovation Seed Funding Award (Grant 1402-2018-2509). K.J.T. was supported by a Department of Education Graduate Assistance in Areas

of National Need Fellowship. E.W.I. was supported by funds from the North Carolina State University Research Experience for Undergraduates, Provost's Professional Experience Programs, and NCSU startup funds.

## Author contributions

K.J.T., J.M.T., and A.J.K. conceived the study. K.J.T., E.W.I., and A.J.K. developed, performed, and analyzed the wet lab experiments. K.V. and J.M.T. developed and performed the software, simulations, file design, and next-generation sequencing analysis with guidance from all. K.J.T. and A.J.K. wrote the paper with input from all.

## Competing interests

K.J.T., J.M.T., and A.J.K. are co-founders of DNAli Data Technologies, Inc. which has licensed rights to PCT Application No. PCT/US2019/049170 titled "Non-Destructively Storing, Accessing, and Editing Information Using Nucleic Acids" filed on August 30, 2019 by North Carolina State University on this work. The remaining authors declare no competing interests.
