## [Peer Review File · Nature Communications]

Reviewers' Comments:

Reviewer #1:

Remarks to the Author:

This manuscript describes the use of mismatched PCR primers in information retrieval from DNA-based data storage systems as a means of implementing a "File preview" operation for files stored in DNA storage. The basic idea is that a subset of the sequences encoding a file are accessed via a fully complementary primer, and the remainder are accessed via mismatched primers that only hybridize and prime efficiently in more promiscuous PCR conditions. The authors show that PCR conditions can be tuned to determine whether only the fully complementary binding sites are amplified, or whether the mismatched sites are also amplified, as a way of determining whether the "full file" or just the "preview" is accessed.

The work is somewhat interesting and the results are compelling, and demonstrate that the system largely works as anticipated. The use of mismatched primers is of potential utility because it could enable more of the sequence space to be utilized than in a system which requires a greater separation between the sequences of the different primers used (Hamming distance of 10 in 20nt primers is used here as the example of a separation between primer sequences that gives maximum orthogonality).

As a minor point of terminology, in Mac OS the "Preview" application opens the full version of files - the authors might be thinking of the "Quick Look" feature. There is also an analogy to image compression techniques such as "Progressive JPEG" in which a low-resolution version of an image can be loaded first (e.g., over a slow Internet connection) with subsequent data loading progressively higher-resolution versions of that image. However, the analogy is not perfect as there is no way under the proposed scheme to obtain the "higher resolution" separately from the "preview" data.

My main criticism of this manuscript is that the system is not very well motivated up-front, or at all really. There is a section near the end of the results section that discusses a hypothetical situation of wanting to search through a database of unknown files for a particular one; in that situation, the "File Preview" operation could lead to lower sequencing costs as only a smaller subset of the strands would need to be sequenced to find the file of interest. However, this is perhaps a somewhat contrived example, at least given the current proposed applications of DNA for long-term data storage and the relatively large cost of accessing files - it seems unlikely that users of DNA data storage would "forget" their files and need to search for them in this manner. Furthermore, given that the same amount of experimental work is required to execute the PCR reaction in either case, meaning that any cost savings are somewhat compensated elsewhere by not getting as much information out of the PCR. In any case, the authors would do well to motivate their work up-front rather than leaving this justification until most of the results have already been presented.

The authors also run into a potential issue of this approach: by utilizing multiple similar primers to access different subsets of a file, and using more promiscuous PCR conditions, they run the risk of mispriming due to sufficiently similar codewords being included in their data of interest. Indeed, this happened in some of the experiments reported here. A claimed advantage of the system proposed here is that more of the available primer sequence space can be utilized compared to the situation where only primers with Hamming distance of 10 or more are used. However, the authors do not seem to consider that using more promiscuous PCR conditions also worsens potential issues with mispriming and likely imposes additional stringent conditions on data encoding to ensure that such issues do not arise. While elegant, the proposed system is "playing with fire" to some extent, and the authors would do well to acknowledge this.

On the whole, the manuscript is well written, other than issues with regard to motivating the work, as outlined above. However, in the first paragraph there are some vague statements about the

advantages of DNA data storage that "arise from the biomolecular structure of DNA which differentiate it from conventional storage media". While this is the first paragraph, this wording could perhaps be made more specific, so as to better inform the reader about the actual advantages of DNA data storage systems and specifically how they arise from the DNA molecular structure.

Reviewer #2:

Remarks to the Author:

The manuscript by Tomek et al. titled "Promiscuous Molecules for Smarter File Operations in DNA-based Data Storage" proposes a new feature for DNA-based data storage systems named "Preview", which is similar to Mac OS Preview function. The method relies on adding a subset of information-coding oligos to an original pool which encodes the file in a smaller size and can be used for easier file access.

The idea is overall interesting. The manuscript is quite well-written, except that in some parts the sentences become too long which makes the content hard to follow and some parts are too concise and need more elaboration. The methods and references are sufficient.

I have some comments/questions and a few concerns about how useful the idea is in terms of addressing the challenges in the field.

1- In the sentence: "Here we show that what were previously considered undesired interactions between DNA molecules can actually be leveraged through thermodynamic tuning to implement novel and useful data access...". I find this phrasing a bit misleading. The part "what were previously considered undesired interactions" can carry the meaning that such interactions were of no use in the past. This is while many techniques rely on using such interactions for creating new sequences (site-directed mutagenesis as an example). I think this sentence should be rephrased and become more specific.

2- In the same page: "... it is this disordered physical nature of DNA storage systems that confer them orders of magnitude greater theoretical information densities over electronic storage systems...". Can authors elaborate on how this disordered-ness gives DNA such higher density?

3- The authors have used the length 20nt for the address sequences and have done their analysis and comparisons based on HD numbers for this length. Since this length is not fixed and address/primers can be of various lengths, I think the authors should emphasize more that the results are for length 20nt.

4- How do the authors financially justify synthesizing additional strands (even a small portion of the original file) to add the Preview feature for searching? For example, I think in case someone wants to find the right file among 15 different files, it would make more sense to add unique "label" sequences to every file (sort of an address) to be able to find the file, instead of recording the whole file in a smaller size. So, it is not that without Preview, one would have to read all the 15 files to pick the right one. I know that Preview gives an idea of how the complete file looks like (similar to Mac's Preview) but this comes with an additional cost. I think the idea becomes more and more useful when dealing with very large sizes. The authors should discuss that.

5- I couldn't find details about the file sizes in the main text. I think it is better if the authors mention that file by file.

We thank the reviewers for their constructive comments. We believe we were able to address them, and that they improved the quality of the manuscript. Reviewer comments are in black, and our description of changes made to the manuscript are in green here and in the manuscript itself.

Reviewer #1 (Remarks to the Author):

This manuscript describes the use of mismatched PCR primers in information retrieval from DNA-based data storage systems as a means of implementing a "File preview" operation for files stored in DNA storage. The basic idea is that a subset of the sequences encoding a file are accessed via a fully complementary primer, and the remainder are accessed via mismatched primers that only hybridize and prime efficiently in more promiscuous PCR conditions. The authors show that PCR conditions can be tuned to determine whether only the fully complementary binding sites are amplified, or whether the mismatched sites are also amplified, as a way of determining whether the "full file" or just the "preview" is accessed.

The work is somewhat interesting and the results are compelling, and demonstrate that the system largely works as anticipated. The use of mismatched primers is of potential utility because it could enable more of the sequence space to be utilized than in a system which requires a greater separation between the sequences of the different primers used (Hamming distance of 10 in 20nt primers is used here as the example of a separation between primer sequences that gives maximum orthogonality).

We appreciate the reviewer's summary and positive feedback. As clarification, multiple primers are not used for accessing different parts of a single file. All subsets of a file are accessed using the same primer pair, while DNA strands encoding different subsets of a file have address sequences that have either perfect matches or different numbers of mismatched nucleotides to the primer pair. Thus, more of the sequence space is used by allowing the addresses sequences on the data strands to be mismatched to the primer, while the sequences of the primers used to access distinct files are able to be kept at highly orthogonal Hamming distances from all other primers. The only variable changed when accessing more or fewer subsets of a file is the promiscuity of the PCR conditions. **We have edited the manuscript text in a few places to clarify this.**

We also included new experimental results that may make this clearer for readers. This is shown in a new panel and caption for Figure 3g, and new text in the results section. As a brief summary here, we have shown that File Preview is capable of a gradient of access conditions, not just of a binary switch of Preview or Full File. We were able to access an intermediate resolution image using an intermediate promiscuous PCR condition. Specifically, a gradient of distinct conditions were identified that were able to specifically access 0, 0-2, 0-2-4, and 0-2-4-5 HD strands as well as successfully decode these low, intermediate, and higher resolution images.

1 - As a minor point of terminology, in Mac OS the "Preview" application opens the full version of files - the authors might be thinking of the "Quick Look" feature. There is also an analogy to image compression techniques such as "Progressive JPEG" in which a low-resolution version of an image can be loaded first (e.g., over a slow Internet connection) with subsequent data loading progressively higher-resolution versions of that image. However, the analogy is not perfect as there is no way under the proposed scheme to obtain the "higher resolution" separately from the "preview" data.

We thank the reviewer for pointing this out. We have edited our analogy as follows: "...an operation with closest analogy to "Quick Look," a function found on modern Mac operating systems or "Progressive JPEG"..." Interestingly, the analogy may be closest to Progressive JPEG in that the DNA containing the 'preview' data is not physically replicated in the higher-resolution version. Rather, promiscuous access conditions capture both preview (0 HD) strands as well as the rest of the strands of the file (>0 HD)

simultaneously to yield the higher-resolution version. In other words, the data contained in the preview strands are not duplicated or redundant with the data in the rest of the >0 HD strands.

2 - My main criticism of this manuscript is that the system is not very well motivated up-front, or at all really. There is a section near the end of the results section that discusses a hypothetical situation of wanting to search through a database of unknown files for a particular one; in that situation, the "File Preview" operation could lead to lower sequencing costs as only a smaller subset of the strands would need to be sequenced to find the file of interest. However, this is perhaps a somewhat contrived example, at least given the current proposed applications of DNA for long-term data storage and the relatively large cost of accessing files - it seems unlikely that users of DNA data storage would "forget" their files and need to search for them in this manner. Furthermore, given that the same amount of experimental work is required to execute the PCR reaction in either case, meaning that any cost savings are somewhat compensated elsewhere by not getting as much information out of the PCR. In any case, the authors would do well to motivate their work up-front rather than leaving this justification until most of the results have already been presented.

We thank the reviewer for their constructive feedback. To better motivate the work up-front, and to better organize the concepts in the introduction, **we modified the first paragraph to be a standalone Abstract. We then completely rewrote the Introduction to provide better motivation up-front.**

3 - The authors also run into a potential issue of this approach: by utilizing multiple similar primers to access different subsets of a file, and using more promiscuous PCR conditions, they run the risk of mispriming due to sufficiently similar codewords being included in their data of interest. Indeed, this happened in some of the experiments reported here. A claimed advantage of the system proposed here is that more of the available primer sequence space can be utilized compared to the situation where only primers with Hamming distance of 10 or more are used. However, the authors do not seem to consider that using more promiscuous PCR conditions also worsens potential issues with mispriming and likely imposes additional stringent conditions on data encoding to ensure that such issues do not arise. While elegant, the proposed system is "playing with fire" to some extent, and the authors would do well to acknowledge this.

We appreciate the reviewer bringing up this important point. As described above in the first response, we initially clarify that multiple primers are not used for accessing different subsets of each file. Accessing all subsets (preview, intermediate, full access) of a file are accomplished using the same primer pair (one primer pair per file). Only the environmental conditions are changed, and the different subsets of DNA strands have addresses with different HD mismatches to the primer pair. Therefore, in this respect, the Hamming distance between primers can be maintained at 10 HD without sacrificing encoding density, and the highest likelihood of mis-priming will be to DNA strands within the same file. The inadvertent inclusion of 5 HD sequences in the data payload in one of our experiments was an error that can be avoided in the future without additional encoding stringency that would normally be used in a 10 HD system. **We have included text to address this in the second Discussion paragraph:**

"...when two particular codewords were adjacent to each other their sequences combined to create a binding site 5 HD from one of the accessing primers. While this was unintended, similar sequences can be avoided in the encoding process using thorough quality control measures that screen through all possible codeword combinations. Primer sequences are typically designed to be more than 8 HD from data payloads; accordingly, we expect data encoding densities can remain unchanged when implementing File Preview since only a single primer pair is used per file."

However, there is an additional consideration as the reviewer mentions. With the use of more promiscuous PCR conditions, mis-priming may occur within the data payload regions even if all data payload regions are set more than 10 HD from all primers. **We have included new text addressing the importance of**

considering and engineering around this possibility in a completely new (3rd) Discussion paragraph. In particular, we discuss that: this is a possibility that should be investigated further in the future in extreme-scale systems; that mispriming events in the preview system are most likely to occur to addresses of strands of the same file that one is accessing which would have minimal effect on the accuracy of file access; that thermodynamically-driven binding interactions are competitive and the presence of addresses < 10 HD from the access primer will generally out compete those from the broader data pool; and that we did not observe off-target access from the randomized 1.5 GB background data in Figure 3h-i.

4 - On the whole, the manuscript is well written, other than issues with regard to motivating the work, as outlined above. However, in the first paragraph there are some vague statements about the advantages of DNA data storage that "arise from the biomolecular structure of DNA which differentiate it from conventional storage media". While this is the first paragraph, this wording could perhaps be made more specific, so as to better inform the reader about the actual advantages of DNA data storage systems and specifically how they arise from the DNA molecular structure.

We thank the reviewer for pointing out this ambiguous point for the audience. **New first and second introduction paragraphs have been written**, per our response to comment #2 above.

Reviewer #2 (Remarks to the Author):

1 - The manuscript by Tomek et al. titled "Promiscuous Molecules for Smarter File Operations in DNA-based Data Storage" proposes a new feature for DNA-based data storage systems named "Preview", which is similar to Mac OS Preview function. The method relies on adding a subset of information-coding oligos to an original pool which encodes the file in a smaller size and can be used for easier file access. The idea is overall interesting.

We thank the reviewer for their summary and positive feedback.

As a quick clarification, no additional strands are required to be synthesized for the Preview functionality. The strands being accessed for a File Preview are not a smaller version of the file, but a small portion (we envision typically ~ <5%) of the whole file of interest. Files are designed to allow for a small subset of the strands to encode the blurry portion of the file while the other subsets encode more of the complete file without repeating what was already encoded in the Preview portion (e.g., 5%, 8%, 15%, and 72%, totaling 100%, of the file is encoded in each subset). To access the small File Preview, one would access the small subset of strands which encode the blurry parts of the image, not by accessing a discrete version of the file. To increase the clarity and color of the image that is accessed, additional subsets of strands are captured in the PCR reaction along with the Preview strands, not by accessing larger versions of the same file. The specific accessing of increasing numbers of subset of strands is controlled by thermodynamic tuning, and the fact that each subset has address sequences that are increasingly different from the accessing primer. **We have edited the manuscript text in a few places to clarify this.**

2 - The manuscript is quite well-written, except that in some parts the sentences become too long which makes the content hard to follow and some parts are too concise and need more elaboration. The methods and references are sufficient.

We thank the reviewer for their constructive feedback. **Sentences throughout the manuscript have been revised and shortened to improve clarity for the reader.**

I have some comments/questions and a few concerns about how useful the idea is in terms of addressing the challenges in the field.

3 - In the sentence: “Here we show that what were previously considered undesired interactions between DNA molecules can actually be leveraged through thermodynamic tuning to implement novel and useful data access...”. I find this phrasing a bit misleading. The part “what were previously considered undesired interactions” can carry the meaning that such interactions were of no use in the past. This is while many techniques rely on using such interactions for creating new sequences (site-directed mutagenesis as an example). I think this sentence should be rephrased and become more specific.

We appreciate the reviewer pointing out this confusing sentence and important biomolecular technique. **We have subsequently condensed this paragraph to be a stand-alone Abstract and have removed the language in question.**

We have also **rewritten the opening of the now 4th paragraph of the introduction**, which contained similar language, to better convey the intent and to draw attention to the use and utility of non-specific interactions in molecular biology:

“We hypothesize that so called non-specific interactions in DNA-based data storage systems, conventionally viewed as a thermodynamic hinderance, can actually be leveraged to expand file address space, increase data storage capacity, and implement novel in-storage functions in DNA storage systems. This hypothesis is inspired by intentional non-specific interactions that have been leveraged for DNA editing in molecular biology (e.g., site-directed mutagenesis) and more recently in DNA storage for in-storage search.”

4 - In the same page: “... it is this disordered physical nature of DNA storage systems that confer them orders of magnitude greater theoretical information densities over electronic storage systems...”. Can authors elaborate on how this disordered-ness gives DNA such higher density?

We have revised that section, now located at the beginning of the second introduction paragraph:

“Organizing, accessing, and finding information constitutes a complex class of challenges. This complexity arises from how information is commonly stored in DNA-based systems: as many distinct and disordered DNA molecules free floating in dense mutual proximity. This has two major implications...”

5 - The authors have used the length 20nt for the address sequences and have done their analysis and comparisons based on HD numbers for this length. Since this length is not fixed and address/primers can be of various lengths, I think the authors should emphasize more that the results are for length 20nt.

We have added the primer length in multiple locations throughout the manuscript to emphasize that the results are for length 20 nt.

6 - How do the authors financially justify synthesizing additional strands (even a small portion of the original file) to add the Preview feature for searching? For example, I think in case someone wants to find the right file among 15 different files, it would make more sense to add unique “label” sequences to every file (sort of an address) to be able to find the file, instead of recording the whole file in a smaller size. So, it is not that without Preview, one would have to read all the 15 files to pick the right one. I know that Preview gives an idea of how the complete file looks like (similar to Mac’s Preview) but this comes with an additional cost. I think the idea becomes more and more useful when dealing with very large sizes. The authors should discuss that.

Please see the clarification above in response 1 regarding how no additional strands are synthesized to implement the Preview feature. As no additional strands are required to be synthesized for File Preview, there is no increased cost for implementation. The only difference in the strands is that the address sequences have small HD differences ('mutations') to the primer sequence, depending on the subset of the file that each strand encodes.

The idea of adding unique 'label' sequences or metadata to improve the ability to search for data is definitely a viable approach. We believe it could be used in conjunction with Preview, particularly in situations where the amount of metadata or specific details of a file is not enough to adequately inform a user which file they are looking for; this would definitely become more and more useful with larger files that contain more overall data. We've also added text to the 2nd paragraph of the introduction to discuss this:

"...while the inclusion of metadata in the strands of DNA could facilitate search, ultimately there will be many situations in which multiple candidate files contain very similar information. For example, one might wish to retrieve a specific image of the Wright brothers and their first flight, but it would be difficult to include enough metadata to distinguish the multiple images of the Wright brothers as they all fit very similar search criteria."

7 - I couldn't find details about the file sizes in the main text. I think it is better if the authors mention that file by file.

We thank the reviewers for this helpful suggestion and have added the file sizes to the main body of the manuscript (6th paragraph of the Results).

"We asked if this tunability could be applied to entire files (NCSU WufLab logo – 25.6 kB, two Wright glider images – 27.9 and 30.9 kB – Figure 3f left and right, respectively, Earth – 27.2 kB) rather than just toy DNA strands."

Reviewers' Comments:

Reviewer #1:

Remarks to the Author:

The authors have responded well to the reviews and added important clarifications and new data to the text that significantly enhance the manuscript. In particular, the clarification that it is only the PCR conditions and not the primers that change is a very important clarification. The additional discussion of the possibility of additional mispriming in large datasets is also welcome. As such I would recommend accepting the revised manuscript.

Reviewer #2:

Remarks to the Author:

I thank the authors for providing the information and improving the text. I think the manuscript can be published in the revised format.